# Local incomplete combustion emissions define the PM$_{2.5}$ oxidative potential in Northern India

Deepika Bhattu [1,18] ✉, Sachchida Nand Tripathi [2,3] ✉, Himadri Sekhar Bhowmik [2], Vaios Moschos [1], Chuan Ping Lee [1], Martin Rauber [4,5], Gary Salazar [4,5], Gülcin Abbaszade[6], Tianqu Cui[1], Jay G. Slowik[1], Pawan Vats[7], Suneeti Mishra[2], Vipul Lalchandani [2], Rangu Satish [8,19], Pragati Rai[1], Roberto Casotto[1], Anna Tobler [1,20], Varun Kumar [1,21], Yufang Hao [1], Lu Qi[1], Peeyush Khare [1], Manousos Ioannis Manousakas [1], Qiyuan Wang[9], Yuemei Han [9], Jie Tian[9], Sophie Darfeuil [10], Mari Cruz Minguillon [11], Christoph Hueglin [12], Sébastien Conil [13], Neeraj Rastogi [8], Atul Kumar Srivastava[14], Dilip Ganguly [7], Sasa Bjelic [15], Francesco Canonaco [1,20], Jürgen Schnelle-Kreis [6], Pamela A. Dominutti [10], Jean-Luc Jaffrezo[10], Sönke Szidat [4,5], Yang Chen[16], Junji Cao[17], Urs Baltensperger [1], Gaëlle Uzu [10], Kaspar R. Daellenbach [1], Imad El Haddad [1] ✉ & André S. H. Prévôt [1] ✉

The oxidative potential (OP) of particulate matter (PM) is a major driver of PM-associated health effects. In India, the emission sources defining PM-OP, and their local/regional nature, are yet to be established. Here, to address this gap we determine the geographical origin, sources of PM, and its OP at five Indo-Gangetic Plain sites inside and outside Delhi. Our findings reveal that although uniformly high PM concentrations are recorded across the entire region, local emission sources and formation processes dominate PM pollution. Specifically, ammonium chloride, and organic aerosols (OA) from traffic exhaust, residential heating, and oxidation of unsaturated vapors from fossil fuels are the dominant PM sources inside Delhi. Ammonium sulfate and nitrate, and secondary OA from biomass burning vapors, are produced outside Delhi. Nevertheless, PM-OP is overwhelmingly driven by OA from incomplete combustion of biomass and fossil fuels, including traffic. These findings suggest that addressing local inefficient combustion processes can effectively mitigate PM health exposure in northern India.

Air pollution is the fifth leading cause of mortality[1] causing ~8 million premature deaths per year globally[2]. Like other economically and demographically growing megacities, New Delhi experiences persistently elevated pollution levels which have adverse effects on human health, regional ecosystems, crop yields, and local climate. The levels of fine particulate matter (PM$_{2.5}$, mass of particles with aerodynamic diameter less than 2.5 μm) in New Delhi are 30 times higher than World Health Organization guidelines 2021[3]. It is estimated that exposure to high levels of ambient PM is responsible for 0.6–1.3 million premature deaths in India each year[4–8], along with 14–33 million years of life lost

(YLL)[5]. Without any intervention, these numbers are expected to increase by 50% by 2030[9]. The severity of the problem has been realized amid the haze events of November 2016, when the Indian government shut almost 6000 schools and completely banned heavy-duty trucks from entering New Delhi[10]. Furthermore, the government has launched a national-level 5-year strategic plan (National Clean Air Program (NCAP), MoEF&CC) in 112 cities across the country to reduce ambient PM$_{2.5}$ concentrations by 20–30% by 2024[11,12]. However, for mitigation strategies to be effective, there is an urgent need to identify and quantify the most significant PM$_{2.5}$ sources, their local vs. regional contributions and particulate toxicities to assess the health impacts.

In-situ measurements of PM$_{2.5}$ chemical composition[13,14], remotely measured atmospheric data[15,16] and air-quality modelling results[17] indicate that residential combustion, traffic, and anthropogenic dust are the most important contributors to PM pollution in India. However, there is still a lack of quantitative understanding of the spatial distribution of these emissions and their contributions to PM-related adverse health effects. Previous source apportionment studies have focused on individual locations in the Delhi National Capital Region (NCR) or a few cities in the Indo-Gangetic Plain (IGP)[14], and the regionality i.e., geographic origin of the sources has never been investigated. While a few recent studies in Delhi have identified the sources of primary OA, the sources of secondary OA, which accounts for half of the OA mass concentration[18–21], have yet to be identified due to the extensive molecular fragmentation used in conventional in-situ analytical techniques such as electron impact ionization and thermal desorption techniques. In this regard, near-molecular level speciation of OA via "soft" ionization techniques is critical to filling this knowledge gap.

Furthermore, molecular-level speciation of OA could facilitate identification of sources controlling the OP of PM$_{2.5}$, a well-known metric for assessing the acute health effects, which remains largely unexplored in this region[22]. While OP studies focused on individual PM components (e.g., organics and transition metals) exist, none of them has targeted the complete PM composition[23,24].

In this study, we identified the most important sources of PM$_{2.5}$ and their OP in the Trans- and Upper-IGP and determined the effects of local vs. regional emissions on their concentrations. We analyzed PM$_{2.5}$ filter samples (cold period: Jan-Mar, warm period: Apr-May) collected from five representative sites in Northern India; two within Delhi (urban background and urban roadside), two encompassing Delhi on its distant north-west (rural background) and adjacent south-east directions (sub-urban industrial), and the distant (~500 km) downwind sub-urban Kanpur site in the east (Fig. 1a). The analysis involved measuring the major aerosol components, including organics, inorganic ions, and trace elements in their oxide form to reconstruct PM$_{2.5}$ mass. We characterized the near-molecular and bulk chemical composition of OA and identified its major primary and secondary source contributors. Further, we measured the magnitude of PM$_{2.5}$-OP using three different acellular assays and identified its main driving sources (Fig. 1b). Finally, we compared PM$_{2.5}$ mass and its OP across sites from Asia-Pacific (India and China) and European (Spain, France, and Switzerland) countries where OP was measured using the same protocol. Our results will be useful for (a) improving emission inventories in future studies, (b) designing evidence-based targeted local and regional control strategies to achieve the NCAP goals, and (c) epidemiological evaluations to improve understanding on relationship between PM and human health.

## Results

### PM$_{2.5}$ composition
Average reconstructed PM$_{2.5}$ concentrations of 98 µg m$^{-3}$ (±39 µg m$^{-3}$) and 40 µg m$^{-3}$ (±16 µg m$^{-3}$) are measured across the five sites during the cold and warm seasons, respectively, slightly higher than winter levels in Chinese urban areas, and up to a factor of 10 higher than those

encountered in winters of European cities (Supplementary Table 1). The spatially homogeneous PM$_{2.5}$ concentrations (Fig. 2a, b) are in sharp contrast to the variability of its constituents, underscoring the importance of local aerosol emissions and formation processes (Figs. 2c–n, 3a–j).

Secondary inorganic aerosol (SIA) constitutes ~29 ± 8% (4–77 µg m$^{-3}$) of PM$_{2.5}$ (Fig. 1b), with ~3 times higher mass concentrations in the cold period. While ammonium sulfate dominates during the warm season, ammonium chloride and ammonium nitrate are mainly observed in colder nights, when low temperatures and high relative humidity drive their partitioning into the particle phase. Ammonium sulfate appears to be regional, with similar concentrations and temporal variability across all sites, in contrast to ammonium chloride and ammonium nitrate which exhibit strong site-to-site differences. Ammonium nitrate formation appears to be more favored outside Delhi, probably due to the inhibition of nitric acid formation by the high NO concentrations inside Delhi during nighttime and suppression of OH due to high VOC concentrations during daytime. In contrast, ammonium chloride, which has been recently identified as a major driver of particle growth in the IGP[25], is particularly important inside Delhi, suggesting the presence of local sources of hydrogen chloride. Trace elements (Cu, Cd, Sn, Sb and Pb) primarily from industries and open waste incineration contribute on average 0.4% to PM$_{2.5}$. Their sources dominate at the Delhi sites (Fig. 2k, l) because of the prevailing east and north-west winds from the surrounding industrial areas[26].

Carbonaceous aerosols constitute more than half of the reconstructed PM$_{2.5}$ mass across all sites and seasons, with an OA:EC ratio of 8 (Fig. 1b). EC is dominated (~70–75%) by fossil fuel emissions, most likely from vehicular exhaust, with two times higher concentrations inside Delhi (Fig. 2m, n). In contrast, the fossil fraction of OA is only ~32% in the cold period, and 40% in the warm season (Fig. 1c).

We identified five major sources of OA, including two primary biofuel combustion sources related to high N-containing fuel i.e., cowdung (termed cold-season primary OA, CPOA) and wood and agricultural waste burning (termed biomass burning OA, BBOA), and one primary OA from traffic exhaust (termed hydrocarbon-like OA, HOA) (Fig. 3). The two remaining sources are dominated by secondary organics formed through oxidation in the atmosphere: cold-season oxygenated OA (COOA) and urban oxygenated OA (UOOA). Near-molecular level (i.e., molecular ion formulae) chemical fingerprints of individual OA components measured by an extractive electrospray ionization long-time-of-flight mass spectrometer (EESI-LToF-MS) are shown in Fig. 4, highlighting the contrasting differences in the complex composition of the identified sources. The molecular characteristics of the organic components, their fossil vs. non-fossil origins, seasonal variability, and local vs. regional behaviors are discussed in the following subsections.

### Organic aerosol sources
Hydrocarbon-like OA (HOA) originates from fresh vehicular tailpipe emissions. The highest average HOA concentrations of 8 µg m$^{-3}$ are recorded at the urban roadside site in Delhi. Similar to EC, HOA concentrations have minimal seasonal variation (Fig. 3i, j). Consistent with previous studies[19], HOA contributes 10–20% of total OA mass with higher relative contributions in the warm season, up to 40% at the urban roadside, representing 50% of the total fossil OA.

Biomass burning OA (BBOA) is characterized by the abundance of anhydrous sugars (C$_6$H$_{10}$O$_5$, Fig. 4a)[27] emitted from the pyrolysis of cellulose in biomass. BBOA night-time concentrations are up to 5-10 times higher during the colder season (6 ± 5 µg m$^{-3}$) due to local residential heating and domestic cooking. Nevertheless, BBOA concentrations remain high during April and May (1 ± 2 µg m$^{-3}$), with clear contribution from open burning of crop residues. This is confirmed by high levoglucosan/mannosan and low levoglucosan/K$^+$ ratios in April-

May shown in Supplementary Fig. 1. Because of its local sources, the absolute concentrations of BBOA exhibit significant spatial variation across the study area (see Fig. 3g, h). BBOA is predominantly non-fossil (98%) and contributes 6 ± 4% of the total OA mass with higher relative contribution in the colder period (up to 23%).

Cold-season primary OA (CPOA) is provisionally attributed to combusting biomass with high N-containing compounds. It had been shown that cow-dung combustion, for instance, commonly done in low-income households for heating and cooking in India, may contain such species[28]. This fraction has been identified based on the predominance of $C_nH_{2n-2}N_2$ molecules (n = 6-10, Fig. 4b, c), which is consistent with the high abundance of N-containing volatile and non-volatile compounds reported in these emissions[28–30]. CPOA is strongly enhanced during the night and exhibits a rather spatially homogeneous contribution (site average: 3–5 μg m$^{-3}$, ~6–9% of total OA). Its concentrations during cold weather are up to 10 times higher (Fig. 3e, f) than during warmer weather, due to the increase in residential heating or cooking emissions and shallower boundary layer

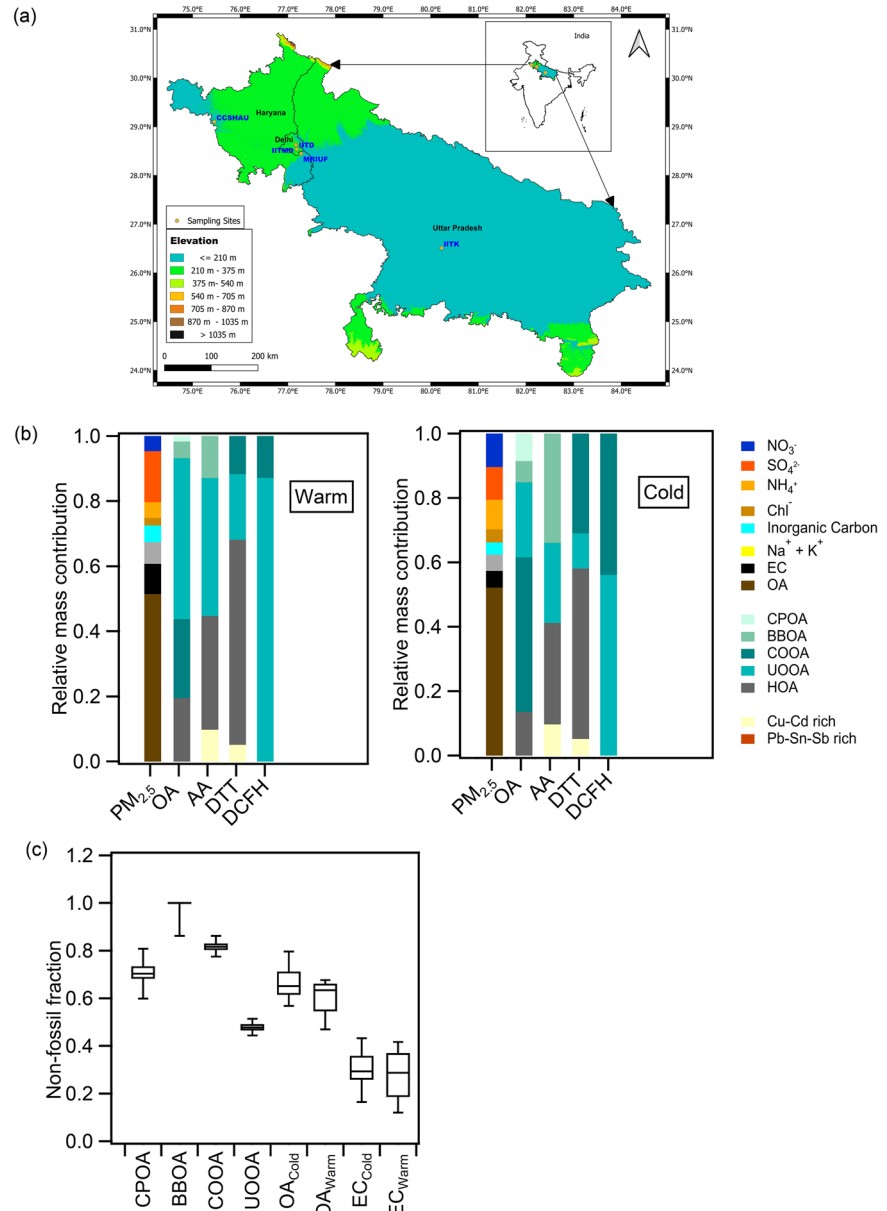

**Fig. 1 | Summary of fine particulate matter (PM$_{2.5}$) chemical composition and source contributions. a** Map of Northern India and sampling site locations: Upwind to downwind – Chaudhary Charan Singh Haryana Agricultural University (CCSHAU Hisar, Haryana - rural background), Institute of Tropical Meteorology Delhi (IITM Delhi - urban background), Indian Institute of Technology Delhi (IIT Delhi - urban roadside), Manav Rachna International University, Faridabad (MRIU Faridabad, Haryana - sub-urban industrial) and Indian Institute of Technology Kanpur (IIT Kanpur, Uttar Pradesh - sub-urban). CCSHAU Hisar and IIT Kanpur are outside Delhi, IITM Delhi and IIT Delhi are within Delhi, and MRIU Faridabad is at the Haryana-Delhi border. **b** Season-specific (Cold: Jan-Mar and Warm: Apr-May) spatially averaged relative mass contributions of the chemical components to total reconstructed PM$_{2.5}$ mass, and source (see text for the name of the sources)

contributions to organic aerosol (OA), and OP (ascorbic acid: AA, dithiothreitol: DTT; 2´,7´-dichlorofluorescin: DCFH) per unit air volume. **c** Contributions of non-fossil fraction (f$_{nf}$) to OA sources and seasonal OA and elemental carbon (EC) (box whisker plots [line/Box: median and 25$^{th}$-75$^{th}$ percentile; upper and lower end of whisker: 5$^{th}$ & 95$^{th}$ percentile]). The contributions were calculated by performing multi-linear regression on uncertainty weighted OA sources (derived from positive matrix factorization) and non-fossil organic carbon fraction measured by radiocarbon ($^{14}$C) analysis. The uncertainties are assessed by performing 1000 bootstrap runs. Seasonal contribution to OA and EC was determined from those 44 filters selected for radiocarbon measurements of both total and elemental carbon fraction (see Supplementary Method 5).

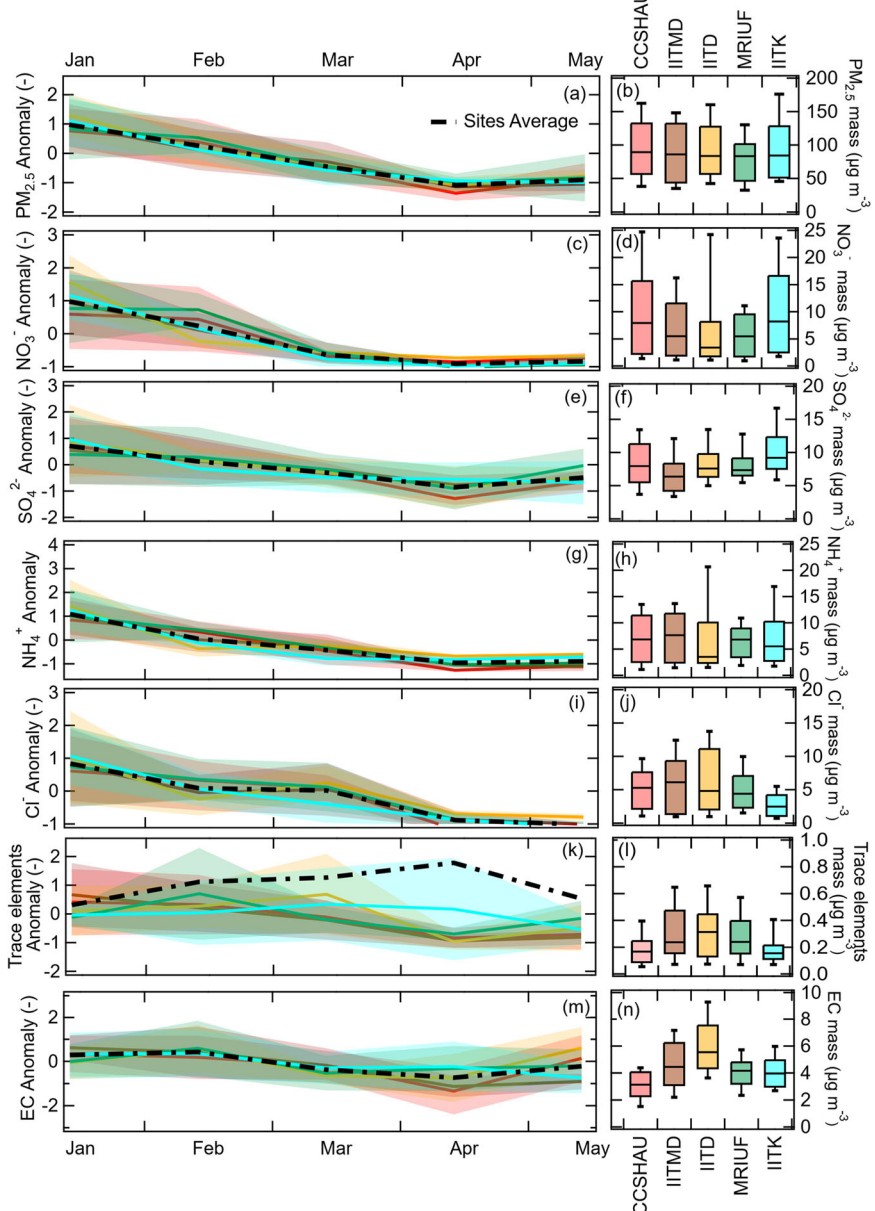

**Fig. 2 | Spatial and temporal variation in fine particulate matter (PM$_{2.5}$) and its species.** Standardized monthly variation of PM$_{2.5}$ and its species except OA for five sites (**a, c, e, g, i, k, m**). The temporal anomaly (y-axis) for each sampling site was calculated from 9-12 (Jan), 8-9 (Feb), 5-6 (Mar), 5 (Apr), and 5 (May) daily samples as: ((absolute value − temporal site average ($\mu$))/temporal site standard deviation ($\sigma$)).

The colored solid lines and corresponding shaded area represent site-specific temporal anomaly and its ±1$\sigma$. A thick black dashed line represents the overall averaged spatial variation. The absolute values of the same parameters are shown as box whisker plots [line/box: median and 25th-75th percentile; upper and lower end of whisker: 10th-90th percentile] in the (**b, d, f, h, j, l, n**).

conditions. Overall, its spatial and temporal variability is similar to that of BBOA, because both fuels are predominant energy sources in Indian households. Both sources are non-fossil (Fig. 1) and correlate well (Pearson's r = 0.5-0.7, n = 140) with incomplete combustion products (e.g., PAHs and C$_5$-C$_{10}$ acids) measured by GC-MS and LC-MS (Supplementary Table 2).

Cold-season oxygenated OA (COOA), a secondary OA dominated by non-fossil sources, is primarily linked to atmospherically processed biomass smoke in the presence of NO$_x$. It is characterized by C$_4$-C$_7$ nitro-aromatics (H:C ∼ 1.3–1.5) and nitro-furans (H:C > 1.5) with 1–2 nitrogen atoms (Fig. 4d), including C$_6$H$_8$N$_2$O$_2$, C$_6$H$_{10}$N$_2$O, C$_4$H$_7$NO$_3$, C$_5$H$_9$NO$_3$, C$_6$H$_{11}$NO$_3$, C$_7$H$_{13}$NO$_2$, and C$_5$H$_{10}$N$_2$O$_2$ (Fig. 4e). In addition, this fraction is composed of non-nitrogen containing CHO compounds with C$_4$-C$_8$ compounds having low H:C ratios, most likely related to oxidation products of aromatic precursors and furans present in

biomass emissions (Fig. 4f). It strongly correlates with the C$_4$-C$_{10}$ dicarboxylic acids measured by LC-MS (Supplementary Table 2). This fraction has negligible day-night variability and shows enhanced concentrations outside Delhi with on average 24 μg m$^{-3}$ (50% of total OA) compared to 15 μg m$^{-3}$ at the Delhi sites (35% of the total OA). Although COOA is dominated by cold season BB emissions (Fig. 3a, b), similar to BBOA, high COOA concentrations are still recorded at the end of April due to agricultural waste burning activities observed during the farmland clearing period.

Urban oxygenated OA (UOOA) is affected by both fossil emissions from vehicle exhausts and non-fossil emissions from cooking (Fig. 4g, h and i). The contribution of traffic emissions is identified by oxidation products of aromatics (e.g., 1,3,5-trimethylbenzene, toluene and naphthalene) and long-chain alkanes (like n-dodecane) having 12-18 C-atoms with 3-5 O-atoms[31]. Cooking emissions within this fraction are

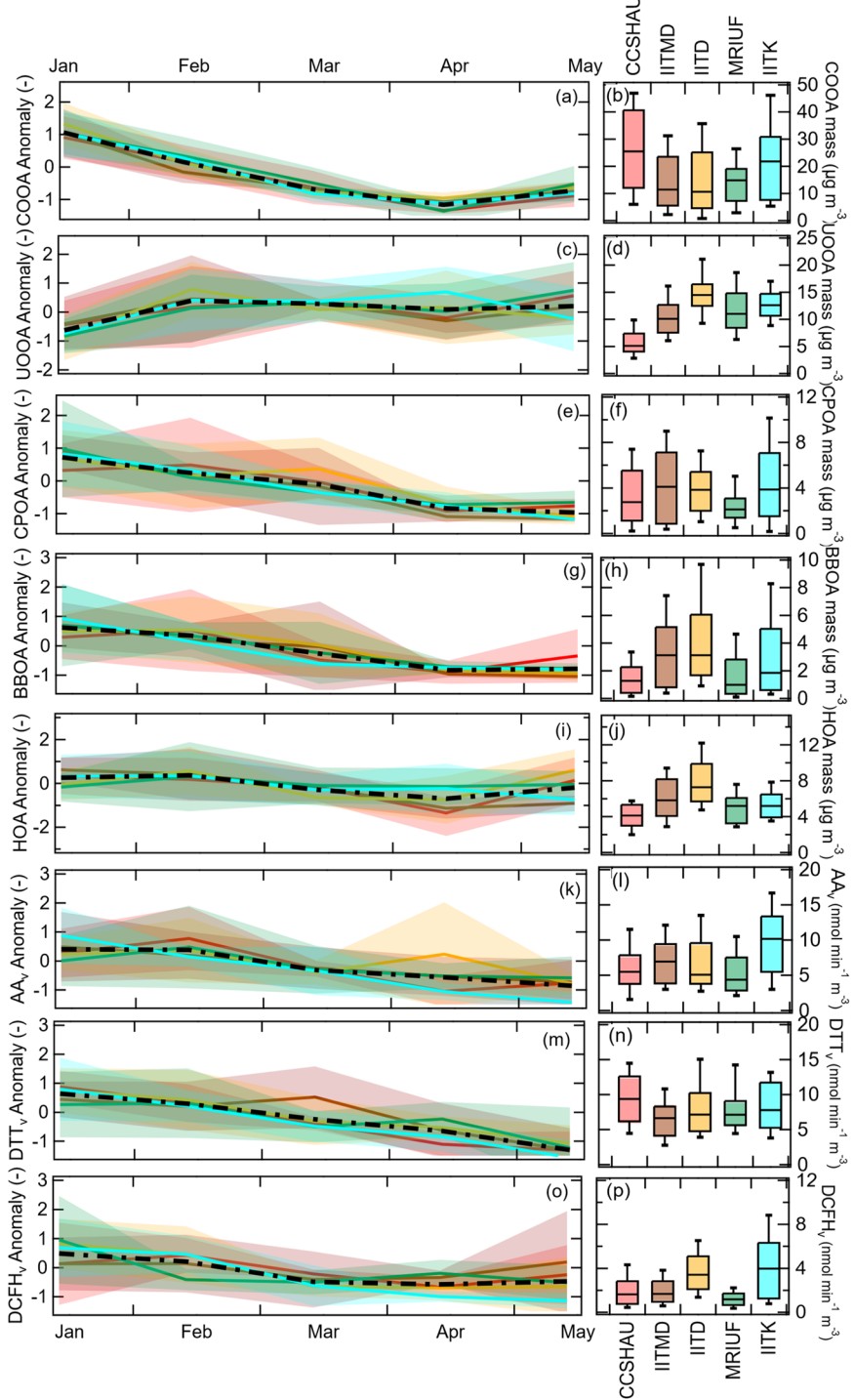

**Fig. 3 | Spatial and temporal variation in organic aerosol (OA) sources and oxidative potential (OP).** Standardized monthly variation of OA sources and OP per unit volume of air, $OP_v$ for five sites (**a, c, e, g, i, k, m, o**). The temporal anomaly (y-axis) for each sampling site was calculated from 9-12 (Jan), 8-9 (Feb), 5-6 (Mar), 5 (Apr), and 5 (May) daily samples as: ((absolute value – temporal site average ($\mu$))/temporal site standard deviation ($\sigma$)). The colored solid lines and corresponding shaded area represent site-specific temporal anomaly and its ±1σ. A thick black dashed line represents the overall averaged spatial variation. The absolute values of the same parameters are shown as box whisker plots [line/box: median and 25th-75th percentile; upper and lower end of whisker: 10th-90th percentile] in (**b, d, f, h, j, l, n, p**).

composed of $C_{16}$-$C_{19}$ unsaturated hydrocarbons (H:C > 1.7, O:C < 0.25)[32]. We note the absence of biogenic emissions even in the warmer period, as suggested by the absence of 3-methyl-1,2,3-butanetricarboxylic acid (3-MBTCA), a higher-generation oxidation product of α-pinene. UOOA has similar concentration levels across seasons and contributes more (1.6-2.5 times) in the warmer period

(Fig. 3c, d). Except for the upwind rural background site (average ~5 µg m⁻³), similar concentrations are found across the sites with slightly higher values at the Delhi urban roadside (average ~15 µg m⁻³).

Overall, OA emissions and their secondary formation processes are local in nature. HOA and UOOA are especially important inside Delhi, while COOA forms outside Delhi. HOA exhibits no seasonal

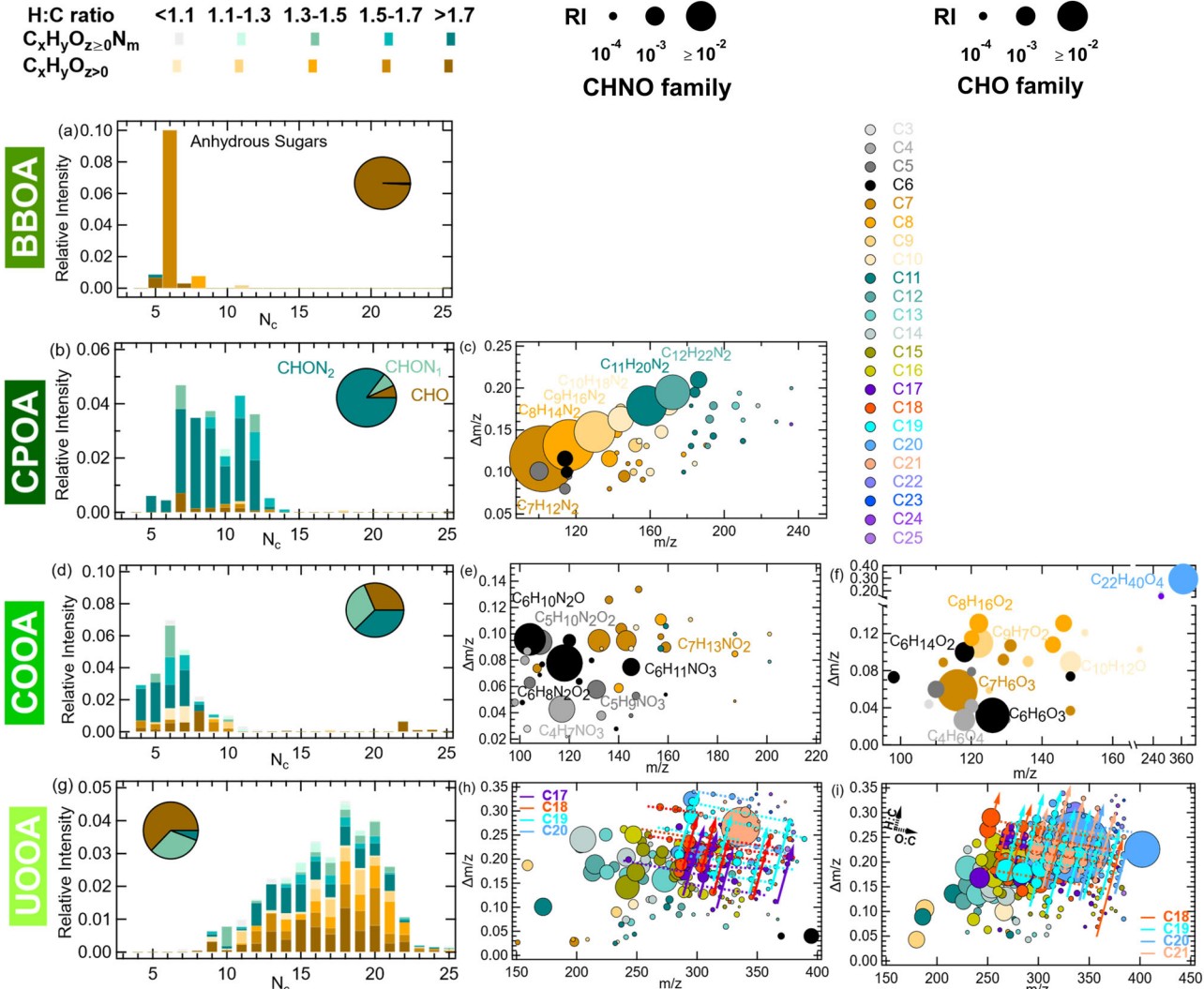

**Fig. 4 | Near-molecular level organic aerosol (OA) source composition.** Only four out of the five identified sources are shown as the HR-EESI-ToF is not sensitive to the water-insoluble hydrocarbon-like OA (HOA). Carbon number distribution (**a, b, d & g**) of biomass burning OA (BBOA), cold-season primary OA (CPOA), cold-season OA (COOA), and urban oxygenated OA (UOOA) [left panels], and their corresponding family-specific ($C_xH_yO_{z>0}$ and $C_xH_yO_{z\geq0}N_m$) mass defect ($\Delta m/z$, difference from integer mass) plots (**c, e, f, h & i**) are shown in the middle and right panels. The relative intensity of $C_xH_yO_{z>0}$ and $C_xH_yO_{z\geq0}N_m$ family species binned according to H:C ratios shows clear differences in the source composition. The

colored circles represent the carbon number ($N_c$), and their size is proportional to the relative intensity of the corresponding species in each OA factor. Every straight dashed line represents a group with species having same # H and C and/or N but different number of oxygen atoms. Formulae for dominant ions in COOA, CPOA and UOOA are annotated in the graph. The relative contribution by species to different compound classes to the total signal intensity illustrates secondary COOA as most oxidized and dominated by both CHO and $CHN_1O$; UOOA as more aromatic and less oxidized, and dominated by CHO. CPOA, on the other hand, is more of aliphatic origin with $CHN_2$ dominance.

variation, while the production rate of the related oxidation products in UOOA are enhanced during summer resulting in its dominant contribution. In contrast, the primary (BBOA and CPOA) and secondary (COOA) fractions related to biofuel combustion are mainly observed during the cold period due to residential heating.

## $PM_{2.5}$ oxidative potential and its sources

To capture the wide variability of reactive oxygen species (ROS) produced during inhalation, three different acellular OP assays (dithiothreitol: DTT, ascorbic acid: AA and 2´,7´-dichlorofluorescin: DCFH) were applied to the samples. We observe 1.5 times higher OP activity outside Delhi when expressed per unit of sampled air ($OP_v$). In the colder season, the activity is 2-4 times higher than in the warm season (Fig. 1b, c) and $OP_v$ dominates at the downwind sub-urban Kanpur site compared to all other sites, except for $DTT_v$ (Fig. 3k–p).

We quantified the intrinsic $OP_m$ (OP per unit aerosol mass; $DTT_m$, $AA_m$, and $DCFH_m$) of the sources of OA and elements using a multiple

linear regression model (Supplementary Table 3) and evaluated the $OP_m$ uncertainty using a bootstrapping technique[33] where input matrices were obtained from the random resampling of the rows from original data to create new matrices having multiple entries of some rows and omittance of other rows. Supplementary Fig. 2 shows the seasonal variations of both $OP_m$ and $OP_v$ for all 3 assays (DCFH, DTT and AA) used in this study. Although different sources contribute differently to the three assays, we find that organics from combustion emissions and their oxidation products dominate the intrinsic oxidative potential of PM with UOOA being the predominant contributor (Fig. 5a). Traffic emissions (HOA) and their oxidation products (UOOA), as well as the oxidation products of biomass burning (COOA) dominate $DDT_m$, potentially because of the large presence of quinones (Supplementary Figs. 3, 4). BBOA is important for $AA_m$, while $DCFH_m$ is dominated by secondary OA (COOA and UOOA). Apart from the OA sources, Cu and Cd (probably from industrial emissions and open waste incineration) contribute ~5-10% to the $OP_v$ together with Pb-Sn-Sb sources (Fig. 5a).

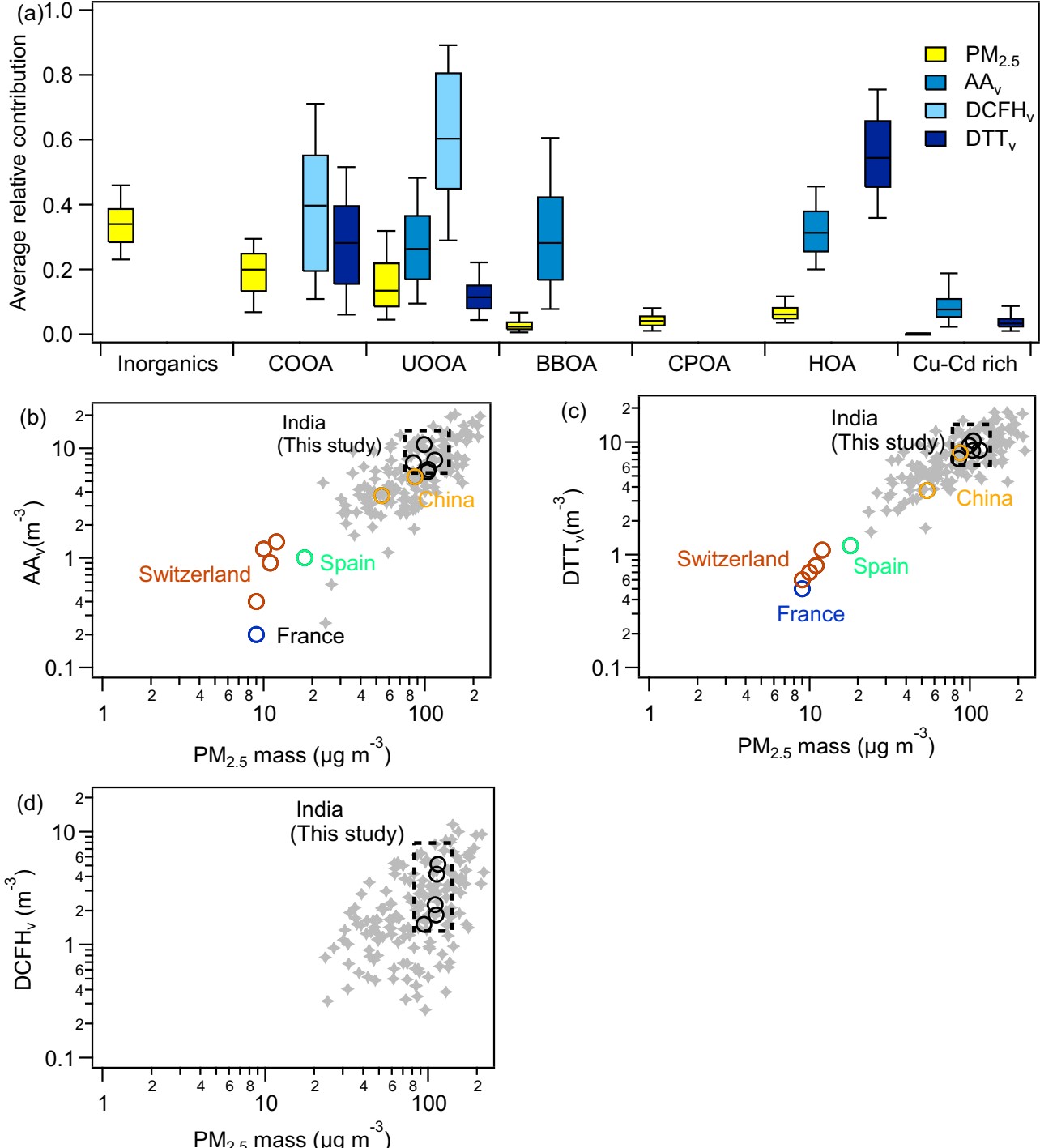

**Fig. 5 | Drivers of fine particulate matter (PM$_{2.5}$) and its oxidative potential (OP$_v$). a** Source contribution to PM$_{2.5}$ mass concentrations ($\mu$g m$^{-3}$) and 3 assays (ascorbic acid: AA$_v$, dithiothreitol: DTT$_v$; 2´,7´-dichlorofluorescin: DCFH$_v$; all in nmol m$^{-3}$). The relative contribution is estimated by considering the median OP per unit mass (OP$_m$) of each assay obtained from 1000 bootstrap runs while performing multi-linear regression, and organic aerosol (OA) sources as well as PM$_{2.5}$ spatial and temporal concentrations. PM$_{2.5}$ ($\mu$g m$^{-3}$) and its oxidative potential comparison across five (5) world regions (India, China, Spain, France, and Switzerland) with distinct economic status (**b**–**d**) indicates ~10 times higher oxidative aerosols in all the Indian locations covered in this study. The samples from these sites were specifically chosen as they are measured with the same protocol and are thus directly comparable.

We find that OP and PM mass are dominated by different emission sources. While secondary inorganic aerosols (SIA) are important for total PM mass, OP is overwhelmingly driven by primary and secondary OA from local combustion emissions (Fig. 5a). Due to contamination, the impact of dust on OP and PM mass could not be estimated, and therefore cannot be excluded.

**Implication for policymaking**

We demonstrate that PM$_{2.5}$ concentrations are exceptionally high throughout the trans- and upper-IGP, but its composition differs substantially due to the local origins of PM components. Despite the spatial variability in PM composition, species driving OP are mainly organic in nature across all sites and seasons. In contrast, in terms of

PM mass concentration, ammonium chloride is a significant contributor within Delhi, whereas ammonium sulfate and ammonium nitrate are dominant outside the city. Cold season $PM_{2.5}$ levels in our study region are a factor of 2 higher than in Chongqing, China, and up to a factor of 10 higher than in European cities e.g., Bure (France), Barcelona (Spain), and Bern, Payerne, Magadino and Zurich-Kaserne (Switzerland) (Fig. 5b–d).

Our findings also reveal that Indian OP outpaces Chinese and European cities by up to a factor of 5, making it one of the highest ever observed in the world (Fig. 5b, c). These high OP levels are primarily created by uncontrolled local incomplete combustion sources, including biofuel and fossil fuel emissions, and their oxidation products.

The air pollution situation in India is alarming, with a growing number of cities experiencing severe pollution despite the implementation of the clean air program. Today, air pollution in India including ambient PM and household pollution, is responsible for 1.67 million deaths every year[6]. Projected demographic shifts in this region indicate that in order to maintain current PM-attributable mortality rates, average PM levels must decrease by approximately 30% within the next 15 years to counterbalance the rise in PM-related deaths resulting from the aging population[34]. Therefore, an effective program to deliver clean air to the Indian population is urgently needed to prevent several million premature deaths every year. Our results suggest that limiting local incomplete combustion of fossil and biofuels will be an effective means to reduce both PM pollution and oxidative potential. While air pollution in India is a nationwide problem, addressing it will require working with local communities and stakeholders to introduce societal changes and raise public awareness of air pollution, thus effectively limiting local incomplete combustion sources.

## Methods
### PM$_{2.5}$ filter sampling
The 12-h (day and night) and 24-h integrated $PM_{2.5}$ quartz fiber filter samples were collected in 2018 using high-volume samplers from January to mid-March and continued with one sample every 3rd day until May at five sites extending from 150 km upwind to 500 km downwind of New Delhi. To understand the effect of local and regional transport of pollutants in the capital city, one rural background upwind site in the north-west of Delhi, two representative sites within Delhi (urban roadside and sub-urban industrial), one bordering Delhi (urban background) and one downwind sub-urban Kanpur site were selected. A total of 330 selected samples were analyzed for chemical composition and oxidative potential over cold (January-March) and warm (April-May) season. More details about the sampling sites and filter collection, storage and transportation are provided in Supplementary Methods 1 and 2.

### Chemical composition analysis
Supplementary Table 4 summarizes the methods for determination of the chemical composition along with auxiliary measurements done in this study.

**Bulk- and near-molecular level organic aerosol (OA).** We determined the bulk- and near-molecular level chemical composition of water-extracted OA using an LToF-AMS (Aerodyne Research Inc., Billerica, MA, USA) and an EESI-LToF-MS (Tofwerk, AG, Switzerland)[35], respectively. The methods hereafter are referred to as offline-AMS and offline-EESI, respectively. The water-soluble filter extracts were prepared[36], where a 16-mm filter punch from each sample was sonicated (20 min at 30 °C) in 10 ml ultrapure water (18.2 MΩ cm, total organic carbon-TOC < 3 ppb) followed by vortexing (1 min) to maintain homogeneity and subsequent filtration using 0.45 μm nylon membrane syringe filters. The resulting aqueous solution was doped with

labelled $^{15}NH_4NO_3$ and $(NH_4)_2^{34}SO_4$ standards (2 ppm each) to track the instrument performance and quantify water-soluble organics (WSOC) present in the ambient atmosphere. Although the relative ionization efficiency (RIE) of labelled ions and ambient organics is different, the assumption is that WSOC scales in proportion to the labelled standards (Supplementary Method 3). Further, the doped filtered solution is re-aerosolized with pure $N_2$ gas (flow rate = 1.3 L min$^{-1}$) using an apex-Q nebulizer (Elemental Scientific, Inc., Omaha, NE, USA) and the produced aerosols were injected simultaneously into the AMS (flow rate = 0.08 L min$^{-1}$) and the EESI (flow rate = 1 L min$^{-1}$). Aerosols injected into the AMS were dried with a Nafion dryer (Perma Pure) whereas the ones injected into the EESI were only diluted (1:3) with pure $N_2$ gas to avoid depletion of primary spray ion signal and then passed through a charcoal denuder to eliminate artefacts due to semi-volatile organics. Overall, the sample was measured for 12 minutes followed by 18 minutes of an ultrapure water blank. To determine the interference from the preceding sample traces in the sampling lines to the EESI, both samples and water blanks were intermittently switched to a HEPA filter at regular intervals (Supplementary Fig. 5). Details on instrument setup, measurements and data analysis are provided in Supplementary Method 4. The ambient aerosol composition ($M_{diff}$) in both the AMS and EESI was calculated by subtracting the preceding signal for water blank ($M_{water\_blank}$) from the sample signal ($M_{sample}$) for all the fitted HR organic fragments (Eq. 1) and the uncertainties were calculated by error propagation. For the EESI, additional data from sample-to-HEPA filter switching ($M_{sample\_filter}$ and $M_{water\,blank\_filter}$) in between measurement of samples was removed before subtracting the water blank from the sample signal (Eq. 2).

$$\text{For AMS,} M_{diff} = M_{sample} - M_{water\,blank} \tag{1}$$

$$\text{For EESI,} M_{diff} = M_{sample} - M_{sample\,filter} - M_{water\,blank} - M_{water\,blank\,filter} \tag{2}$$

Further, to apportion the contributions by fossil and non-fossil particulate carbon (total, organic, elemental), carbon-14 ($^{14}C$) in total carbon (TC) was determined on all filter samples and elemental carbon (EC) on 44 selected filters from 3 sites (upwind rural background, Delhi urban background and urban roadside) covering all periods. The $^{14}C$ content of TC was measured using the procedure described in ref. 37 using an elemental analyzer coupled with the accelerator mass spectrometer Mini Carbon Dating System (MICADAS) at the Laboratory for the Analysis of Radiocarbon (LARA; University of Bern, Switzerland)[38,39]. Due to low water solubility and high charring of OC, a modified extraction and desorption temperature protocol (Bern-India_4S) was developed for determination of the $^{14}C$ content of EC. Details on measurement protocol and estimation of the fossil and non-fossil fractions are provided in the Supplementary Method 4.

We measured OC and EC by the EUSAAR-2 thermal-optical transmission method using a Sunset analyzer, WSOC and WSIC using a TOC analyzer, cations ($K^+$, $Na^+$, $NH_4^+$) and anions ($Cl^-$, $NO_3^-$, $SO_4^{2-}$) using ion chromatography and a range of targeted organic compounds (acids, PAHs, oxy-PAHs, anhydrous sugar, resin acids, alkaloids, hopanes, n-alkanes, higher n-alkanes and lignin pyrolysis products) for selected samples (Supplementary Table 4). We analyzed 29 trace elements (Li, Mg, As, Ca, Sc, Ti, V, Cr, Mn, Fe, Co, Ni, Cu, Se, Rb, Sr, Zr, Mo, Pd, Cd, ln, Sn, Sb, Cs, Ba, Ce, Pt, Tl and Pb) using inductively coupled plasma mass spectrometry (see Supplementary Method 4).

### Source apportionment of OA and trace elements
**Positive matrix factorization.** Organic aerosol source apportionment was performed separately on the LToF-AMS and EESI-LToF-MS datasets using positive matrix factorization (PMF)[40] implemented with the Multi-linear Engine (ME-2)[41] and Source Finder (SoFi) package[42] (v.

6.8B). PMF is a bilinear model that linearly resolves the sample matrix (X) into two non-negative matrices; one representing mass spectral profiles (or factors; F) and the other representing time-dependent concentrations of profiles (G) and a residual error matrix (E) using a weighted least-squares approach.

**Bulk OA sources.** The AMS-PMF input consisted of a data matrix composed of time series of 785 HR organic fragment ions and the corresponding error matrix (Total error $s_{i,j}$; Eq. 3) which included measurement uncertainty, $\delta_{i,j}$[43,44] as well as sample and water blank variability (Eq. 4).

$$\text{Total error } s_{i,j} = \sqrt[2]{\delta(blank)_{i,j}^2 + \sigma(blank)_{i,j}^2 + \delta(sample)_{i,j}^2 + \sigma(sample)_{i,j}^2}$$

(3)

$$\sigma(blank or sample)_{i,j} = \frac{\sqrt[2]{\frac{1}{N} * \sum \left(x_{i,j} - \bar{x}_i\right)^2}}{N}$$

(4)

where $N$ is the number of spectra measured for each sample or blank, $x_{i,j}$ is the signal of HR fragment ions in each spectrum and $\bar{x}_i$ is the average signal of HR fragment ions $(j)$ for each sample $(i)$ or blank $(i)$.

Ions with signal-to-noise ratio (SNR) less than 2 were downweighted by a factor of $10$[45]. Initially, only the ions above $m/z$ 44 were used as input to PMF (Supplementary Figs. 6d-f). An unconstrained PMF was initially performed for $n$ (no. of factors) = 3–8 with three random seed runs for each factor (18 runs) for reasonable manual inspection, while later, all HR ions were included. Supplementary Fig. 6 summarizes the PMF diagnostics ($\Delta Q/Q_{exp}$, scaled residuals and factor profiles for different runs) and the basis on which the optimum number of factors were chosen, and the stability of the solution was interpreted. A 6-factor solution was found to be the optimum solution, and it consists of cold-season oxidized OA (COOA), urban OA (UOOA), biomass burning OA (BBOA), cold-season primary OA (CPOA) and two unknown factors (Factor 1a and Factor 1b) (Supplementary Fig. 6g). The latter were identified as sample contamination factors based on their comparable concentrations in samples and field blanks (Supplementary Fig. 6h), while the concentrations of the real factors were close to zero in the field blanks. CPOA in the 6-factor solution was constrained (*a* value of 0.5) with a clean profile from the 7-factor solution because of its high correlation with CPOA derived from unconstrained EESI-PMF (Pearson's $r = 0.7$, $n = 394$). A 7-factor solution resulted in uninterpretable splitting of the contamination-derived factor 1a, while the 5-factor solution had CPOA mixed across BBOA and COOA. Further, 8 + -factor solutions had splitting of COOA and Factor 1a.

To assess the rotational ambiguity and estimate related uncertainties of the optimum 6-factor solution, bootstrapping (BS, 100 runs) was performed with input data (ions $m/z > 44$) and corresponding error matrix, and the results were compared to the base case. The resulting factor time series (100 runs × 6 factors) were used to define the upper and lower concentration limits for each factor, while performing PMF on all HR ions (including the ones below $m/z$ 44). Further, the 6-factor solution (average of 10 seed runs) was found to be stable by comparing the factor profiles and time series of the average base solution (with all HR ions) with 100 BS runs.

Further, we performed 2D hierarchical clustering on the AMS-derived factor profiles to identify unique fragment ions related to a single factor or a group of factors based on the current dataset which otherwise would have been overlooked due to their lower signal intensity and medium to high variability (Supplementary Fig. 6i). For each AMS-derived factor, a specific cluster of closely associated ions was observed and is discussed in Supplementary Method 5.

**HOA estimation.** HOA was not apportioned in water soluble OA due to its lower solubility ($< 10\%$)[36], but rather estimated using approach 1 where $EC_{nf}$ is estimated from levoglucosan and estimated $EC_f$ is compared with HOA measured by AMS/ACSM at two sites (urban roadside and urban background) to estimate HOA at all the remaining sites. In approach 2, the average measured $EC_f/EC_{nf}$ ratio was used instead of levoglucosan. The comprehensive methodology for both approaches, along with the pertinent equations is detailed in Supplementary Method 5. Further, comparison of approach 1 and 2 suggested the suitability of approach 2 in the absence of specific source markers i.e., levoglucosan.

**Species-specific factor recovery.** We used a new approach to obtain factor-specific recoveries, which were used to estimate total factor concentrations and thus both the water-soluble and insoluble organic fractions. Bulk water-insoluble OC (WIOC) was calculated by subtracting bulk WSOC and $HOC_{estimated}$ (available for the same three sites) from the total OC (Eq. 5).

$$WIOC(t) = OC(t) - WSOC(t) - HOC_{estimated}(t)$$

(5)

A new PMF input data matrix was prepared using all the HR fragment ions used in water-soluble AMS PMF together with one additional variable "WIOC" and finally scaled to WSOM using the bulk OM:OC ratio. While performing PMF, the upper and lower bounds of the resulting factor time series were constrained from the BS runs of the base case solution of water-soluble AMS-PMF. Further, to estimate the uncertainty of base case solution obtained from the new PMF input, 100 BS runs were performed on this new data matrix and the stability of runs is shown in Supplementary Fig. 7a. The WSOM fraction ($f_{WSOM,k}$) apportioned for each factor was calculated in each run and the corresponding $WSOC_k$ was determined (Eq. 6). $f_{WSOM_i,k}$ was defined as the ratio of the sum of all variables for each factor except the last variable ($WIOC_{i,k}$) to the sum of all variables including the last variable (Eq. 7).

$$WSOC_k(t) = \left[\frac{f_{t,k} * [WIOC(t) + WSOM(t)]}{\left(\frac{OM}{OC}\right)_k}\right]$$

(6)

Where $f_{t,k}$ is the explained variation for each factor obtained from PMF and $\left(\frac{OM}{OC}\right)_k$ is the water-soluble factor (WSOC$_k$; Eq. 6)

$$f_{WSOM_i,k} = \frac{\sum\limits_{i=1}^{n} WSOM_{i-1,k}}{\sum\limits_{i=1}^{n} WSOM_{i,k}}$$

(7)

Where $k$ is the number of factors and $n$ is the number of total variables in the PMF input.

$$WIOC_k(t) = \left[\frac{f_{t,k} * (1 - f_{WSOM,k}) * [WIOC(t) + WSOM(t)]}{\left(\frac{OM}{OC}\right)_k}\right]$$

(8)

$$R_k(t) = \frac{WSOC_k(t)}{WSOC_k(t) + WIOC_k(t)}$$

(9)

The variability of factor recoveries $R_k$ (Eq. 9) is shown in Supplementary Table 5. Finally, the median value of $R_k(t)$ was used for calculating total $OC_k$ (Eq. 10). CPOA was found to be least water-soluble (53%) compared to UOOA, COOA and BBOA (94–100%). The contamination factors (factor 1a and factor 1b) had solubilities of 88% and

94%, respectively.

$$OC_{i,k} = \frac{WSOC_{i,k}}{R_k} \qquad (10)$$

The mass closure was performed by comparing the reconstructed OC mass (AMS-PMF-based CPOA, BBOA, COOA, and UOOA, and $HOC_{estimated}$) with total OC measured by the Sunset analyzer (Supplementary Fig. 7b). A good agreement ($R^2 = 0.9$ and slope = 1.1) between the two suggests adequacy of the applied method.

**Fossil and Non-fossil fractions of OA sources.** Radiocarbon analysis ($^{14}C$) coupled with offline-AMS source apportionment[46,47] was used to determine the fossil (traffic exhaust and coal burning) and non-fossil fraction (biogenic emissions and biomass burning including heating and open fires of agricultural and other solid wastes) of the primary OA sources as well as the secondary OA sources identified based on their SOA precursors. Apart from unequivocal fossil and non-fossil separation of primary OA sources, this approach corroborates our PMF apportionment method and interpretation of the factors' nature or sources.

To calculate the fossil and non-fossil fractions of each OA source, multiple linear regression (MLR) was performed on the samples ($n = 44$) where the fossil and non-fossil fractions of both TC and EC were determined from $^{14}C$ analysis (Supplementary Table 4) and the fossil fraction of OC ($OC_f$) was calculated. Briefly, two approaches were used to estimate fossil and non-fossil fraction independently from each AMS-derived OA factor using their uncertainty-weighted mass concentrations and compared (Supplementary Table 6). The details of the MLR procedure used in approach 1 and 2 are provided in Supplementary Method 5.

Except $HOC_{est}$, no major primary fossil source was found. UOOC (52% of total fossil) was observed to be the major fossil secondary source. Among the non-fossil fractions, BBOC was the major primary source linked to biomass burning, and COOC and UOOC were the major secondary sources linked to oxidized products of biomass combustion, and of vehicular and cooking emissions, respectively. The spatiotemporally averaged non-fossil contributions of COOA followed by UOOA were found to be highest among all OA sources. The source contributions were calculated by multiplying the AMS-PMF-apportioned OA mass with the source-specific fossil and non-fossil contributions obtained from the multi-linear regression model.

**Near-molecular level composition of OA sources.** The EESI-PMF data and corresponding error matrix contained 1454 ions ($m/z$ 120–444 normalized to the primary ion [NaI]Na$^+$; with positive signal left after subtracting the water blanks from the ambient samples). The error matrix was prepared in a similar way as the AMS-PMF input.

Initially, an unconstrained PMF was performed for 5-10 factors with 2 random seed runs for each factor (12 runs in total) for manual inspection and to evaluate the stability of the factors. A large decrease in $Q/Q_{exp}$ was observed when the number of factors increased from 5 to 6 and thereafter small changes were observed. The 6-factor solution yielded factors identified as BBOA, COOA, CPOA and UOOA, and two contamination factors (factor1a$_{AMS}$ and factor1b$_{AMS}$); all correlated well with the AMS-PMF-derived factors (Supplementary Table 7), however, a cleaner BBOA factor (Pearson's $r$ between BBOA derived from AMS- and EESI-PMF increased from 0.6 to 0.7; without normalization to the labelled ion) is obtained in the 9-factor solution. In the 5-factor solution, BBOA was mixed with COOA whereas in the 7- and 8-factor solutions, physically uninterpretable splitting of BBOA was observed. The time series of absolute concentrations (EESI and AMS time series normalized to labelled Na$_2$$^{34}$SO$_4$, and $^{15}$NO$_3^-$ or $^{34}$SO$_4$$^{2-}$, respectively) are compared (Supplementary Table 7).

We ran PMF again for the 5-and 6-factor solution in a constrained mode by using the reference profile of the cleaner BBOA (from the unconstrained 9-factor solution) with an $a$ of 0.3 (after performing sensitivity analysis over $a = 0$–$0.5$ with a step size of 0.1) and by limiting the upper and lower range of the AMS-derived contamination factor-1b time series obtained from the BS runs (30) with an $a$ of = 0.4. The 6-factor solution was chosen as the "base case" hereafter. The uncertainty of the base case was estimated by performing 100 bootstrap runs.

The averaged optimum 6-factor solution with relative signal intensity and explained variation of ions in the different factors is shown in Supplementary Fig. 8a. The apportioned ions are divided into 5 families: $C_xH_yO_1$, $C_xH_yO_{n>1}$, $C_xH_yN_z$, $C_xH_yO_1N_z$ and $C_xH_yO_{n>1}N_z$. Further, the atomic H:C vs O:C ratio plots of ions belonging to the CHO and CHNO family in COOA and UOOA are shown in Supplementary Figs. 8b, c. Five different groups are formulated based on the modified aromaticity index (AI$_{mod}$)[48,49] and H:C ratio[50] where group 1 represents combustion-derived condensed polycyclic aromatics (AI$_{mod}$ > 0.66), group 2 represents vascular plant-derived polyphenols (0.66 ≥ AI$_{mod}$ > 0.50), group 3 represents highly unsaturated and phenolic compounds (AI$_{mod}$ ≤ 0.50 and H:C < 1.5), group 4 represents aliphatic compounds (2.0 ≥ H:C ≥ 1.5), and group 5 represents saturated fatty and carbohydrates (H:C > 2.0). The COOA signal is dominated by CHNO family ions present in group 3 and 4 suggesting the presence of both aromatic and aliphatic nitro-compounds. On the other hand, the UOOA signal is dominated by CHO family ions present in group 3, 4 and 5 with lower O:C ratio suggesting a mix of aromatic and aliphatic compounds together with saturated fatty acids. Similar to AMS-PMF, CHN ions (# of N-atoms = 2) contributed most to the CPOA and $C_6H_{10}O_5$ dominated BBOA.

In this innovative overall procedure, and despite performing separate PMF on AMS and EESI dataset, we have successfully used EESI source apportionment as a tool to identify near-molecular level chemical fingerprints of the AMS-derived secondary factors related to their origin and/or atmospheric transformations. However, the quantification of the factor contributions was carried out on the AMS dataset.

Further, the details on the identification of sources of trace elements is provided in Supplementary Method 5. Briefly, we identified three important factors i.e., a K$^+$-Na$^+$ rich, a Cu-Cd rich, and a Pb-Sn-Sb rich factor contributing to the elemental mass.

**PM$_{2.5}$ oxidative potential (OP)**

We determined the oxidative potential both the volume-normalized OP (OP$_v$; nmol m$^{-3}$) and the mass-normalized OP (OP$_m$; nmol μg$^{-1}$) of PM$_{2.5}$ for the three acellular assays performed (DTT, AA and DFCH). Details of the measurement protocol of all three assays are discussed in Supplementary Method 4. The associations of OP$_v$ with total PM$_{2.5}$, its constituents and their sources are displayed in a correlation matrix (Supplementary Table 8). To identify the main compositional drivers for each assay response, a stepwise linear regression model was used (Eq. 11).

$$OP_v(t) = \sum_i OP_{m,OA_i} * OA_i(t) + \sum_i OP_{m,traceelements_i} * TE_i(t) \qquad (11)$$

Where $OA_i$ and $TE_i$ are defined as OA and trace elements source concentrations, respectively. $OP_{m,OA_i}$ and $OP_{m,trace\ elements_i}$ are defined as OA and trace elements source strength or their OP activity per unit mass, respectively. The details of the model are further discussed in Supplementary Method 5. The contribution of both elemental and OA contamination factors was subtracted from the final predicted OP. The model uncertainty was assessed by bootstrapping (1000 BS runs). The comparison of modelled source-apportioned OP$_v$ and measured OP$_v$ (Pearson's $r = 0.8$ (AA$_v$); 0.6 (DCFH$_v$ and DTT$_v$)) suggests close

agreement within the uncertainties for the 3 assays (Supplementary Fig. 3).

## Data availability
All the data used in this study is openly available in the Dryad repository, accessible at https://doi.org/10.5061/dryad.280gb5mvx. The shapefile for the GIS-based map (Fig. 1a) is taken from freely available data source (https://gadm.org/) and plotted in open platform QGIS software. Source data are provided as a Source Data file. Please contact the corresponding authors when using the data. Source data are provided in this paper.

## Code availability
Datalystica Ltd is the official distributor of SoFi Pro licenses. Igor was used for graph plotting.

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

## Acknowledgements
We thank for the financial support of the Swiss Development Cooperation (SDC) Clean Air Project in India (grant no. 7F-10093.01.04) and the SDC Clean-Air-China Program (7F-09802.01.03). A.S.H.P. also acknowledge the Sino-Swiss Science and Technology Cooperation (SSSTC) HAZECHINA (IZLCZ2_169986). S.N.T. acknowledges the support of Central Pollution Control Board, Government of India, and Centre of Excellence (ATMAN) approved by the office of the Principal Scientific Officer to the Government of India. S.N.T. also acknowledges support under J.C. Bose National Fellowship under the aegis of Science and Engineering Research Board, Department of Science and Technology, Government of India. The support from philanthropies including Bloomberg Philanthropies and the Open Philanthropy, and the Clean Air Fund is also gratefully acknowledged. I.E.-H. acknowledges the support from Ultra-high resolution mass spectrometer, Orbitrap-MS – Project number: 206021_198140 – SNF, R'equip. The chemical analysis on the Air-O-Sol facility at IGE was made possible with the funding received by J.-L.J. for some of the equipment by the Labex OSUG@2020 (ANR10 LABX56). He also acknowledges the funding from the French Research Agency - ANR (GetOPstandOP ANR-19-CE34-0002), Predictair- FUGA Grant, ACME (ANR-15-IDEX-02) for OP measurements at IGE. N.R. acknowledges the support from Department of Space, Government of India. K.R.D. acknowledges support by the Swiss National Science Foundation Ambizione grant PZPGP2_201992. We thank the financial contribution of the Swiss National Science Foundation for the project MOLORG (200020_188624).

## Author contributions
A.S.H.P. and S.N.T. designed the study. D.B. and I.E.-H. analyzed the data and wrote the initial manuscript. P.V., R.S., V.L., S.M., P.R. and V.K. collected on-site filter samples. D.B., C.P.L., R.C., T.C., and J.G.S. performed the offline-AMS and -EESI-ToF analysis. H.S.B. performed EC, OC, TOC, IC, and ICP-MS analysis. G.U., S.D., P.A.D., and J.-L.J. performed all OP (AA, DCFH and DTT) measurements. V.M., M.R., G.S., and S.S. performed $^{14}C$ measurements and V.M. conducted the data analysis. G.A. and J.S.-K. performed the GC-MS analysis. S.B. performed the Orbitrap analysis. K.R.D. and F.C. provided codes for offline analysis and SoFi. Q.W., Y.H., J.T., Y.C, L.Q., M.I.M., J.C. contributed to the setup, measurements, and analysis of data in China while Yuf.H., L.Q., P.K., M.I.M., A.T., J.G.S, N.R., A.K.S., and D.G. contributed to the setup, measurements, and analysis of data in India. M.C.M, C.H., and S.C. contributed to the particulate matter data for sites from Switzerland, Spain, and France, respectively. U.B. contributed to the scientific discussion. All co-authors reviewed and commented on the manuscript.

## Competing interests
The authors declare no competing interests.

## Additional information

[1]Laboratory of Atmospheric Chemistry, Paul Scherrer Institute, Villigen PSI, Switzerland. [2]Department of Civil Engineering & Department of Sustainable Energy Engineering, Indian Institute of Technology Kanpur, Uttar Pradesh, India. [3]Department of Sustainable Energy Engineering, Indian Institute of Technology Kanpur, Uttar Pradesh, India. [4]Department of Chemistry, Biochemistry and Pharmaceutical Sciences, University of Bern, Bern, Switzerland. [5]Oeschger Centre for Climate Change Research, University of Bern, Bern, Switzerland. [6]Comprehensive Molecular Analytics (CMA), Department Environmental Health, Helmholtz Zentrum München, Neuherberg, Germany. [7]Centre for Atmospheric Sciences, Indian Institute of Technology Delhi, New Delhi, India. [8]Geosciences Division, Physical Research Laboratory, Ahmedabad, India. [9]Institute of Earth Environment, Chinese Academy of Sciences, Xi'an, China. [10]University Grenoble Alpes, IRD, CNRS, INRAE, Grenoble INP*, IGE (Institute of Environmental Geosciences), Grenoble, France. [11]Institute of Environmental Assessment and Water Research (IDAEA-CSIC), Barcelona, Spain. [12]Laboratory for Air Pollution and Environmental Technology, Swiss Federal Laboratories for Materials Science and Technology (Empa), Duebendorf, Switzerland. [13]ANDRA DRD/GES Observatoire Pérenne de l'Environnement, Bure, France. [14]Indian Institute of Tropical Meteorology, Ministry of Earth Sciences, New Delhi, India. [15]Biogenergy and Catalysis Laboratory, Paul Scherrer Institute, Villigen PSI, Switzerland. [16]Chongqing Institute of Green and Intelligent Technology, Chinese Academy of Sciences, 400714 Chongqing, China. [17]Institute of Atmospheric Physics, Chinese Academy of Sciences, Beijing, China. [18]Present address: Department of Civil and Infrastructure Engineering, Indian Institute of Technology Jodhpur, Rajasthan, India. [19]Present address: College of Engineering, Science, Technology and Agriculture, Central State University, Wilberforce, Ohio, USA. [20]Present address: Datalystica Ltd., Park innoAARE, Villigen, Switzerland. [21]Present address: Department of Environmental science, Aarhus University, Roskilde, Denmark. ✉e-mail: dbhattu@iitj.ac.in; snt@iitk.ac.in; imad.el-haddad@psi.ch; andre.prevot@psi.ch

