## [Peer Review File - NEW · Nature Communications]

Local incomplete combustion emissions define the PM2.5 oxidative potential in Northern IndiaReviewer #1 (Remarks to the Author):

This research provides rich data and information through sampling, experiments, and data analysis. The conclusion that "addressing local inefficient combustion processes can effectively mitigate PM health exposure in northern India" provides valuable insights for identifying the sources of PM-OP in northern India.

I have a few minor suggestions for your consideration:

1) In the manuscript, some of the figures that display important results are located in the supplementary material, while the main text contains only three sets of figures. It may be worth considering whether some of these figures with significant conclusions could be moved to the main text to enhance the presentation of the findings.

2) line 413 and Supplementary Materials line 271/282: Why were only 15 and 20/30 bootstrap runs used here, whereas other sections typically use 100 or 1000? Could you further explain the rationale behind the choice of the number of bootstrap runs?

Reviewer #2 (Remarks to the Author):

Comments

The importance of the results on the oxidative potential of the samples is unquestionable. The results show the importance of evaluating air quality and the need to reduce emissions in India. Also unquestionable are the sampling procedures in different sites and the huge number of analyses that demanded time and methodologies developed for the immense number of species determined.

Key results

The concentrations are extremely high, as expected and seen in many previous studies. This once again reveals the urgency of lowering the levels to the ones recommended by the WHO.

The number of samples collected in the cold season is large (line 280), but the number collected in the warm season (March-May) seems much lower. In the line 281, it is written that 2 samples were collected a month (biweekly) in the warm season.

On the other hand, in the Supplementary method 2 (line 51) one filter every 6th sampling day (until March) and 1 filter from every 3rd day (rest of the period, warm season) were selected for analysis.

If this is so, although the importance of individual sample results, it is difficult to compare the two seasons: different number of samples in different periods.

The selected samples descriptions are confusing, please comment on this and rewrite. In fact, how many samples were selected to be analysed in each season and, can they be compared?

In this direction, the Supplementary Figure 1 must be more detailed in the text. The Figure give us the idea that the same number of samples were analysed in every month.

Figure 1b: season specific, describes the relative mass contribution of the species and source contributions to OA and OP. The country is big, calculating an average concentration of different locations and draw conclusions; it seems that authors are generalizing. It is likely that are peculiarities at each site and perhaps the authors could select some interesting results (or samples) to discuss individually.

Figure 1c. Contribution of non-fossil fraction to OA. The same situation above, the results are generalized for samples collected in different sites.

It is interesting to observe the Figure 3, the urban organic aerosol (UOOA) showed higher to oxidative potential using DCFHv assay.

Supplementary material

Line 109, F- 81 organic markers. Author generalized, not all PAH, for example, are markers. In fact, many of these species are emitted by different sources. See definition of markers in: Eganhouse, R.P., 2004 (The Geochemical Society Special Publication vol 9, p.143-158; Padoan et al., 2020 (Environmental Research vol 186, 109587) and specially the many papers by Berndt Simoneit and collaborators (Atmospheric Environment vol. 41, p. 8183-8204 (2007).

Supplementary Figure 1. Spatial and temporal variation in PM2.5 species. The way it is presented, it seems that the selected samples are representative of the month. It is not real because authors

selected different number of samples, in different months. Maybe, it's better to display in the figure, the date of the samples (like Figure 5, Supp Mat).

Figure 3 – Supp. Mat. Very comprehensive figure: it explains the molecular diversity in biomass burning, distinguishing the type of biofuels.

Figure 4 – Supp. Mat. Again, the seasonal variation. How to compare different number of samples selected in different seasons. Please, comment on this.

We would like to thank the reviewers for their comments and suggestions to improve the manuscript. The reviewer comments are in black, our responses in blue, and modification to the manuscript in blue and italics.

We trust that the incorporated changes have duly addressed the reviewer's concerns and contributed to the overall clarity and rigor of the manuscript.

REVIEWER COMMENTS

Reviewer #1 (Remarks to the Author):

This research provides rich data and information through sampling, experiments, and data analysis. The conclusion that "addressing local inefficient combustion processes can effectively mitigate PM health exposure in northern India" provides valuable insights for identifying the sources of PM-OP in northern India.

Response: We thank the reviewer for their positive and encouraging feedback that has allowed us to further improve the manuscript.

I have a few minor suggestions for your consideration:

Comment 1. In the manuscript, some of the figures that display important results are located in the supplementary material, while the main text contains only three sets of figures. It may be worth considering whether some of these figures with significant conclusions could be moved to the main text to enhance the presentation of the findings.

Response: We thank the reviewer for the valuable suggestion. In response to that, we have now moved supplementary Figure 1 to the main text as Figure 2 considering its frequent reference and discussion in the main text.

Comment 2. line 413 and Supplementary Materials line 271/282: Why were only 15 and 20/30 bootstrap runs used here, whereas other sections typically use 100 or 1000? Could you further explain the rationale behind the choice of the number of bootstrap runs?

Response: We thank the reviewer for pointing this out. There is a typo in the number of bootstraps runs in line 271 and 282 of Supplementary Information (SI). We have actually performed 100 BS runs and not 20/30 runs. We have now corrected them in the revised SI text. The same numbers were already reported in the main text in line 379 and 384.

There are several factors on which number of bootstrap runs depend such as dataset size, computational resources, stability of estimates, precision requirement as well as best practices in the field. We acknowledge the significance of higher number of bootstrap runs in case of less number of samples and small parameter space. However, in our study, we have dealt with large number of samples (>300 with repetitions) and large parameter space (785 ions for AMS-PMF and 1454 ions for EESI-PMF). Additionally, we have followed EPA recommended number of bootstrap runs (i.e. 100) for performing PMF (EPA PMF 3.0 User guide). We would also like to mention that the number of runs were also restricted by computational limitations (total run time for 100 runs is 28 mins and 317 mins for AMS and EESI, respectively).

Further, we show the stability of the bootstrap runs with coefficient of variation (CoV, ratio of standard deviation to the average of 100 runs) for all factors in Figure 1 presented below. The CoV values for “line 282 of SI, where BS runs were done on all HR ions (including the ones below m/z 44) after constraining the solution with upper and lower concentration limits for each factor)” are within the acceptable range of <20%. We have now included this Figure1 in SI as Fig. S1h.

Figure 1: Box and whisker plot (10th, 25th, 50th, 75th and 90th percentile) of the CoV obtained for the optimum AMS-PMF solution where 100 BS runs were performed on all HR ions (including the ones below m/z 44) after constraining the solution with upper and lower concentration limits for each factor.

In the main text line 413, 15 bootstrap runs (on data matrix of water-soluble AMS ions and WIOC) were conducted because we already constrained the PMF solution using upper and lower concentration limits for each factor. Following the reviewer’s suggestion, we extended the number of bootstrap runs from 15 to 100 and noted that the solutions remained stable (CoV < 20%) with minimal alterations. The higher coefficient of variation (CoV) for BBOA is observed in the runs where the sample concentration decreases by a factor of 100 (Figure 2).

Figure 2: Box and whisker plot (10th, 25th, 50th, 75th and 90th percentile) of the CoV obtained for the optimum PMF solution used to estimate species-specific factor recoveries. 100 BS runs were performed on water-soluble AMS ions and WIOC while constraining the solution with upper and lower concentration limits for each factor.

We have included Figure 2 in SI as Fig S7a and changed the number of bootstrap runs from 15 to 100 in the main text. Line 413 now reads as “Further, to estimate the uncertainty of base case solution obtained from the new PMF input, 100 BS runs were performed on this new data matrix and the stability of runs is shown in Supplementary Fig. 7a.”

Reviewer #2 (Remarks to the Author):

Comments

The importance of the results on the oxidative potential of the samples is unquestionable. The results show the importance of evaluating air quality and the need to reduce emissions in India. Also unquestionable are the sampling procedures in different sites and the huge number of analyses that demanded time and methodologies developed for the immense number of species determined.

Response: We thank the reviewer for their positive and encouraging feedback that has allowed us to further improve the manuscript.

Key results

Comment 1. The concentrations are extremely high, as expected and seen in many previous studies. This once again reveals the urgency of lowering the levels to the ones recommended by the WHO.

The number of samples collected in the cold season is large (line 280), but the number collected in the warm season (March-May) seems much lower. In the line 281, it is written that 2 samples were collected a month (biweekly) in the warm season. On the other hand, in the Supplementary method 2 (line 51) one filter every 6th sampling day (until March) and 1 filter from every 3rd day (rest of the period, warm season) were selected for analysis.

If this is so, although the importance of individual sample results, it is difficult to compare the two seasons: different number of samples in different periods. The selected samples descriptions are confusing, please comment on this and rewrite. In fact, how many samples were selected to be analysed in each season and, can they be compared.

Figure 4 – Supp. Mat. Again, the seasonal variation. How to compare different number of samples selected in different seasons. Please, comment on this.

Response: We thank the reviewer for their constructive feedback. We acknowledge that the term “biweekly” has caused confusion regarding the large differences in the sample numbers between cold and warm period. We would like to clarify that we collected ~50 samples (day/night and daily) at each of the five sites in the cold period (50 x 5 sites = 250) and 10 daily samples in the warm period (10 x 5 sites = 50).

For clarification, we have now changed the text to one sample every 3rd day in the main text (line 280-281). The text is now changed from “...continued biweekly until May” to “...continued with one sample every 3rd day until May”.

We would also like to mention that we observed a continuous shift in meteorological parameters (RH & T) and source contributions throughout the measurement campaign. Consequently, we divided the whole period into two parts to facilitate a clearer understanding of the differences during cold and warm periods. In response to your recommendation, we conducted Welch’s t-test on all parameters (shown in now Figure 2) to compare the cold and warm periods. The results are presented in Table 1 below. Welch’s t-test is generally performed when two independent groups have uneven data points and their variances are not assumed to be equal. The lower p-value (<0.01) and higher degrees of freedom (df>40) indicate that the differences between the two periods are statistically significant.

Parameter	Cold period (average)	Warm period (average)	p-value	Degree of freedom
PM _{2.5} (µg m ⁻³)	101.85	40.37	<0.001	203.23
nmol DTT. min ⁻¹ .µg ⁻¹	0.09	0.13	<0.001	43.63
nmol AA min ⁻¹ µg ⁻¹	0.08	0.12	<0.01	44.13
DCFH nmol [H ₂ O ₂] equiv µg ⁻¹	0.03	0.05	<0.01	45.26
nmol DTT. min ⁻¹ .m ⁻³	9.25	4.99	<0.001	96.92
nmol AA min ⁻¹ m ⁻³	8.22	4.65	<0.001	66.03
DCFH nmol [H ₂ O ₂] equiv m ⁻³	1.75	3.16	<0.001	123.60
CPOA (µg m ⁻³)	2.35	1.04	<0.001	164.98
BBOA (µg m ⁻³)	0.82	0.67	<0.001	148.09
COOA (µg m ⁻³)	23.30	5.98	<0.001	165.53
UOOA (µg m ⁻³)	28.34	11.74	0.23	98.89
HOA (µg m ⁻³)	20.42	4.85	<0.01	84.64
EC (µg m ⁻³)	4.68	3.69	<0.01	70.76
NO ₃ ⁻ (µg m ⁻³)	11.75	1.78	<0.001	273.60
Cl ⁻ (µg m ⁻³)	7.08	1.47	<0.001	283.82
NH ₄ ⁺ (µg m ⁻³)	9.76	2.15	<0.001	285.44
SO ₄ ²⁻ (µg m ⁻³)	9.85	5.89	<0.001	122.18

Comment 2. In this direction, the Supplementary Figure 1 must be more detailed in the text. The Figure give us the idea that the same number of samples were analysed in every month.

Response: Thank you for the comment. We acknowledge the need for additional details regarding the sample count across different months and sites. In the revised Supplementary Figure 1, we have included the number of samples in the figure caption and moved it to main text as Figure 2. For each sampling site, 32-36 daily samples were used. Specifically, the distribution of samples from January to May for all sites is as follows: 9-12, 8-9, 5-6, 5, and 5, respectively.

Line 710 of the revised manuscript now states “The temporal anomaly (y-axis)from 9-12 (Jan), 8-9 (Feb), 5-6 (Mar), 5 (Apr), and 5 (May) daily samples...”

Comment 3. Figure 1b: season specific, describes the relative mass contribution of the species and source contributions to OA and OP. The country is big, calculating an

average concentration of different locations and draw conclusions; it seems that authors are generalizing. It is likely that there are peculiarities at each site and perhaps the authors could select some interesting results (or samples) to discuss individually.

Response: We agree with the reviewer that averaging concentration across all locations may lead to the loss of valuable information. Nevertheless, we observed that the PM_{2.5} concentrations exhibited uniformity across sites and shared common sources of Organic Aerosols (OA) and Oxidative Potential (OP) but with varying contribution levels. In response to the reviewer's suggestion to highlight peculiarities at each site in terms of site-to-site variability and site-wise seasonal variation, we have moved Supplementary Figure 1 to the main text as Figure 2. Kindly note that the comprehensive discussion already existing in the submitted main text pertaining to site-wise overall composition (mentioned in lines 126-128, 133-140, and 144-147) states that while PM_{2.5} concentration is consistent across sites (Fig. 2a, b), there is significant variability in its constituents (Fig. 2c-x). Ammonium sulfate shows regional consistencies, while ammonium chloride/nitrate exhibit site-specific differences. Ammonium nitrate formation is more prevalent outside Delhi, influenced by elevated NO levels at night inhibiting nitric acid formation. Ammonium chloride, a crucial driver for particle growth, is significant in Delhi, suggesting local hydrogen chloride sources. Additionally, carbonaceous aerosols contribute over half of PM_{2.5} mass, and EC, dominated by fossil fuel emissions, shows notably higher concentrations in Delhi (Fig. 2w, x).

Further, we discussed site- and season-wise differences in individual sources of Organic Aerosols (mentioned in lines 162-167, 170-171, 176-177, 183-187, 200-204, 212-215 and 217-222) stating that fresh vehicular emissions contribute Hydrocarbon-like Organic Aerosol (HOA), with the highest average concentration of 8 µg m⁻³ at Delhi's urban roadside. It represents 10-20% of total OA mass (up to 40% in warm seasons). Biomass Burning OA (BBOA) has higher night-time concentrations in the cold season due to local heating and cooking, contributing non-fossil material (98%) and 6 ± 4% to total OA mass (up to 23% in colder periods). CPOA increases at night and has spatially homogeneous contributions, peaking in cold weather. COOA is dominant outside Delhi, showing cold season biomass burning influence, and UOOA, with consistent levels, contributes more in warmer periods. Overall, OA emissions are local, with HOA and UOOA significant in Delhi, while COOA is prevalent outside. Primary and secondary biofuel-related fractions peak during the cold period.

We also discussed the seasonal and site-wise variability in oxidative potential and contribution of different sources to OP as mentioned in lines 226-229 and 236-238). We observe 1.5 times higher oxidative potential (OP_v) activity outside Delhi. In colder seasons, the activity is 2-4 times higher than in warmer seasons, with OP_v dominating at the downwind suburban Kanpur site compared to all other sites, except for DTTv. Despite various sources contributing differently to the three assays, organics from combustion emissions and their oxidation products, especially UOOA, dominate the intrinsic oxidative potential of PM.

Comment 4. Figure 1c. Contribution of non-fossil fraction to OA. The same situation above, the results are generalized for samples collected in different sites.

Response: We would like to clarify that Figure 1c shows the non-fossil fraction of seasonal total EC and total OA, along with the non-fossil fraction of individual OA

components. For instance, biomass burning OA (BBOA) is primarily non-fossil, whereas urban oxygenated OA (UOOA) is composed of both fossil and non-fossil carbon. In alignment with the reviewer's suggestion, we now demonstrate the site-to-site and site-wise seasonal variability of these OA sources, by relocating Supplementary Figure 1 to the main text as Figure 2. The new figure 2 is extensively discussed in the text.

Comment 5. It is interesting to observe the Figure 3, the urban organic aerosol (UOOA) showed higher to oxidative potential using DCFHv assay.

Response: We concur with the reviewer's observation. It is noteworthy to emphasize that urban oxygenated organic aerosol (UOOA) is the highest contributor to the oxidative potential (OP) as shown in Figure 3. UOOA, comprising of both fossil emissions from vehicle exhausts (including oxidation products of aromatic and long-chain alkane precursors) and non-fossil emissions from cooking (comprising unsaturated hydrocarbons), dominates the oxidative potential for DCFH assay among all the sources. This increased sensitivity is attributed to the assay's responsiveness to the particle-bound organic peroxides. Based on the reviewer comment, we have revised the main text in line 236-2380 as "Although different sources contribute differently to the three assays, we find that organics from combustion emissions and their oxidation products dominate the intrinsic oxidative potential of PM *with UOOA being the predominant contributor especially for the DCFH assay (Fig. 4a).*"

Comment 6. Supplementary material Line 109, F- 81 organic markers. Author generalized, not all PAH, for example, are markers. In fact, many of these species are emitted by different sources. See definition of markers in: Eganhouse, R.P., 2004 (The Geochemical Society Special Publication vol 9, p.143-158; Padoan et al., 2020 (Environmental Research vol 186, 109587) and specially the many papers by Berndt Simoneit and collaborators (Atmospheric Environment vol. 41, p. 8183-8204 (2007).

Response: We acknowledge the reviewer's point that not all measured targeted organic compounds are definitive markers of Organic Aerosol (OA) sources. Accordingly, we have incorporated reviewer's suggestion by replacing the term "organic markers" with "*targeted organic compounds*" in both line 109 and Table S2 and S4 of the supplementary information.

Comment 7. Supplementary Figure 1. Spatial and temporal variation in PM_{2.5} species. The way it is presented, it seems that the selected samples are representative of the month. It is not real because authors selected different number of samples, in different months. Maybe, it's better to display in the figure, the date of the samples (like Figure 5, Supp Mat).

Response: We acknowledge the reviewer's concern regarding representation of monthly variations. However, balancing the comprehensive nature of this multi-site analysis which involves an extensive number of samples, and the necessity for a more effective representation posed a challenge.

In response to the reviewer's recommendation, we are relocating Figure S1 to the main text as Figure 2. Additionally, we have incorporated information on the number of samples in the caption. For each sampling site, a total of 32-36 daily samples were

utilized. Specifically, the total count of samples used from January to May is as follows: 9-12, 8-9, 5-6, 5, and 5, respectively.

Line 710 of the revised manuscript now states “The temporal anomaly (y-axis)from 9-12 (Jan), 8-9 (Feb), 5-6 (Mar), 5 (Apr), and 5 (May) daily samples...”

Furthermore, we would also like to mention that we will share the data set used in the figure preparation via journal’s open access data policy.

Comment 8. Figure 3 – Supp. Mat. Very comprehensive figure: it explains the molecular diversity in biomass burning, distinguishing the type of biofuels.

Response: Thank you for your acknowledgment. We have previously discussed it in submitted main text, specifically, in lines 171-174, stating, “Nevertheless, BBOA concentrations remain high during April and May ($1 \pm 2 \mu\text{g m}^{-3}$), with clear contribution from open burning of crop residues. This is confirmed by high levoglucosan/mannosan and low levoglucosan/K⁺ ratios in April-May shown in Supplementary Fig. 2.”